# Influence of Gold Nanoantennas on the Photoluminescence of Silicon Nanocrystals

Ronja Köthemann *, Christian Golla, Hong Qu and Cedrik Meier

Physics Department & CeOPP, Paderborn University, Warburger Str. 100, 33098 Paderborn, Germany
* Correspondence: ronja.koethemann@upb.de

**Abstract:** We study the influence of gold nanoantennas on the photoluminescence signal of silicon nanocrystals. Unlike bulk silicon, which only exhibits low photoluminescence at room temperature due to its indirect band gap, silicon nanocrystals have the advantage of producing strong and size-dependent photoluminescence. Here, we place gold nanoantennas on a layered system in which silicon nanocrystals are integrated. The nanoantennas are embedded in the layered system by subsequent overgrowth. We find that the photoluminescence signal can be manipulated ranging from attenuation to enhancement. Moreover, we investigate the impact of grating coupling and the number of antennas per antenna array on the amplification of the photoluminescence signal.

**Keywords:** silicon nanocrystals; photoluminescence; plasmonic nanoparticles

## 1. Introduction

Silicon (Si) is one of the most important semiconductor materials in both microelectronics [1–3] and photovoltaics [4–6]. Despite the advantages of silicon, which are that it is cheap and non-toxic and has excellent electronic properties, there is one disadvantage: the light emission of indirect semiconductors is very weak. This problem can be overcome by reducing the size of silicon crystals to a few nanometers, which enables efficient optical emission [7,8] and other interesting properties such as band-gap tailoring [9–11] and the potential for implementation as highly sensitive photodetectors [12].

With the use of optical fibers, photonics has been integrated into telecommunications. However, the data must be converted between electronic and optical signals, which is associated with losses. Therefore, it is of enormous importance to be able to amplify optical signals such as photoluminescence (PL). One approach to this is the use of plasmonic nanoantennas, which can be arranged periodically and vary in size from a few angstroms to about 100 nm [13–17]. In the vicinity of these structures there is strong enhancement of the electric field caused by localized surface plasmon resonances at the particle's dipole resonance wavelength. When this resonance coincides with the photoluminescence of the silicon nanocrystals (NCs), amplification is possible. The greatest photoluminescence enhancement occurs at the nanoantenna tips. Thus, when the antennas are deposited on the surface of the layered system, most of the electric field enhancement is not used. Therefore, nanoantennas embedded in the layered system lead to further enhancement of the optical signal [18].

## 2. Fabrication

The layered system in which the silicon nanocrystals are embedded is grown by plasma-enhanced chemical vapor deposition (PECVD) on a boron-doped Si (111) substrate. Alternating thin films of SiON of about 6 nm thickness and $SiO_2$ of about 2 nm thickness are deposited. During high-temperature treatment in a tube furnace at 1250 °C under a nitrogen atmosphere, the Si atoms in the SiON layers cluster together to form nanocrystals because this layer is not thermally stable (see Figure 1a,b). The maximum diameter of the

nanocrystals is determined by the thickness of the SiON layers. Here, the nanocrystals have an average size of 3.5 nm. The SiO$_2$ layers serve as barrier layers. The exact parameters for the PECVD-based fabrication process can be found in [19].

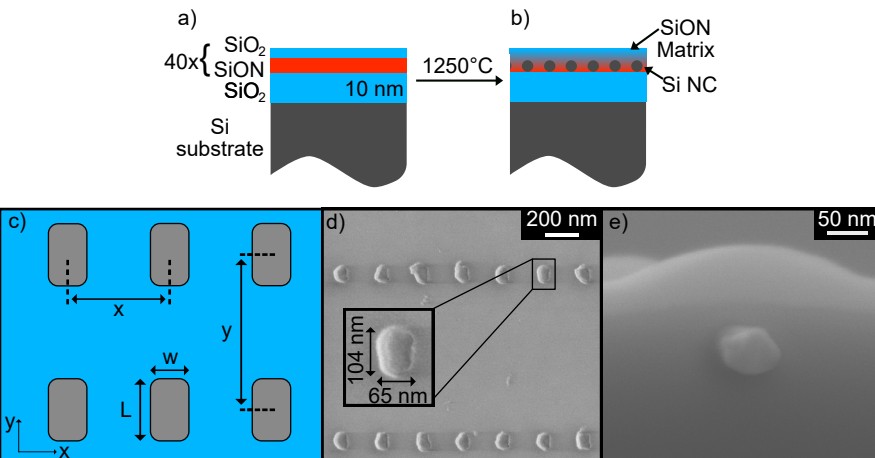

**Figure 1.** (**a**) Schematic sketch of the sample structure including the SiON/SiO$_2$ superlattice. (**b**) Si NCs embedded in a SiON matrix after high-temperature treatment. (**c**) Schematic top view of the antennas on the layered system with the used terms of the parameters. (**d**) Nanoantennas before overgrowth with a length of 100 nm, a width of 80 nm, and a lattice constant of 250 nm (preset sizes of the lithography mask). The inset shows an antenna with the actual measured size dimensions. (**e**) Cross-section of an antenna after overgrowth and annealing.

Gold is used as the material for the antennas as it does not oxidize and has a higher melting point than silver, which is another common material for plasmonic nanoantennas. Moreover, the diffusion of gold in SiO$_2$ is about one order of magnitude lower than that of silver [20]. The nanoantennas are deposited onto the layered system with the silicon nanocrystals. Subsequent growth of another layered system embeds the antennas in the superlattice with nanocrystals.

The antennas are patterned in arrays with a size of 100 µm × 100 µm. In each nanoantenna array, the same geometric parameters are applied. Based on electromagnetic simulations, length, width and lattice constant in the *x*-direction are varied. The distance in the *y*-direction is kept constant at 1 µm. See Figure 1c for a schematic top view of the antennas and the used parameters. The antennas are processed by electron beam lithography with an accelerating voltage of 25 kV onto the approximately 475 nm thick film system. Positive resist CSAR62 is used. Development takes place in AR600-546 and in stopper AR600-60 followed by rinsing in deionized (DI) water. Electron beam evaporation is used to deposit 3 nm of chromium to improve the adhesion of the gold layer. The gold layer is deposited with a thickness of 30 nm. The antennas are finalized after a lift-off process in 80 °C AR600-76 remover followed by rinsing with DI water. The antennas have the largest electric field at their ends, and since the antennas are to amplify the photoluminescence signal from the nanocrystals, another layered system of alternating SiON and SiO$_2$ layers with a thickness of 95 nm is deposited on the antennas. The nanocrystals are again formed in a high-temperature process under a nitrogen atmosphere at 1250 °C. SEM measurements of a small section of an array of non-overgrown gold antennas can be found in Figure 1d with an inset of one antenna and the measured dimensions. In Figure 1e, the cross-section of an antenna after overgrowth and annealing is shown. The size of the antenna is similar to the size before overgrowth, indicating that there has been little diffusion of the gold into the surrounding material. The antenna appears to be rounded, suggesting that the gold has melted into a structure with a smaller surface area. A lower annealing temperature could probably prevent this melting. However, this would lead to the nanocrystals being smaller and having a blue-shifted emission [21].

The gold antennas are so small that they are at the resolution limit of the fabrication process. This leads to slightly different antenna sizes in a single antenna array. The lattice constant can be maintained without deviations.

## 3. Theoretical Results

In order to obtain initial parameters for the size and the lattice constant of the nanoantennas, a theoretical analysis is performed using finite integration technique (FIT). The unit cell for the computation consists of a single nanoantenna on the substrate. Periodic boundary conditions are applied in the $x$- and $y$-directions. This leads to a periodic arrangement of the antennas in an infinite antenna array. The layered system with the silicon nanocrystals in the $SiO_2$/SiON superlattice is used as the substrate for the simulation. The refractive index is determined by ellipsometry, with the layered system approximated as a Cauchy model [22]. A non-dispersive refractive index at a wavelength of 750 nm of 1.49 is used. To show the behavior of the antennas before overgrowth, vacuum is assumed as surrounding material. To model the overgrown case, the refractive index of the layered system is used. For the gold antennas, the optical properties of [23] are used. In the simulations, the height of the antennas is set constant at 30 nm and the spacing of the antennas in the $y$-direction is left at 1 μm. The width is varied between 70 nm and 80 nm, the length between 100 nm and 130 nm, and the distance in the $x$-direction between 200 nm and 700 nm. In the case of resonant excitation, the generated electric field exhibits maxima, especially at the ends of the antennas, which is used to determine the resonance wavelength. The amplitude of the electric field is shown in Figure 2a. In each case, the frequency of the resonance is used, so that a smaller frequency is used for the overgrown antenna. From the distribution, therefore, both cases look similar with a field radiated at the ends of the dipole. By overgrowing the antennas with another layered system containing the nanocrystals, the refractive index of the surrounding material increases. This shifts the resonance to longer wavelengths (see Supplementary Materials).

A larger grating constant and longer antennas also cause a red shift of the resonance. The goal is for the gold antennas in the overgrown case to amplify the photoluminescence from the surrounding silicon nanocrystals, which have their peak intensity between 700 nm and 750 nm. For an antenna length of 130 nm, the resonance is around 875 nm and continues to increase for larger lattice constants (see the upper part of Figure 2). At a smaller antenna length of 100 nm, the resonance shifts to smaller wavelengths, in the range of the PL signal of the nanocrystals. Since the resonance experiences a red shift at larger lattice constants, the overlap with the photoluminescence signal of the nanocrystals also becomes smaller here (Figure 2, the lower part in green). The largest overlap occurs at the smaller lattice constants. The maximum electric field is at a lattice constant of 500 nm for smaller antennas (see Figure 2, the lower part with the black dots) and 550 nm for longer antennas. Normalizing the electric field strength to the antenna density by multiplying the electric field with the number of antennas per antenna array results in the effective electric field becoming larger for smaller lattice constants (see Figure 2, the lower part with the black squares). Thus, despite the largest electric field occuring for larger lattice constants, the choice of smaller lattice constants seems to be more reasonable, since the field calculated based on the number of antennas plus the overlap with the photoluminescence signal is larger.

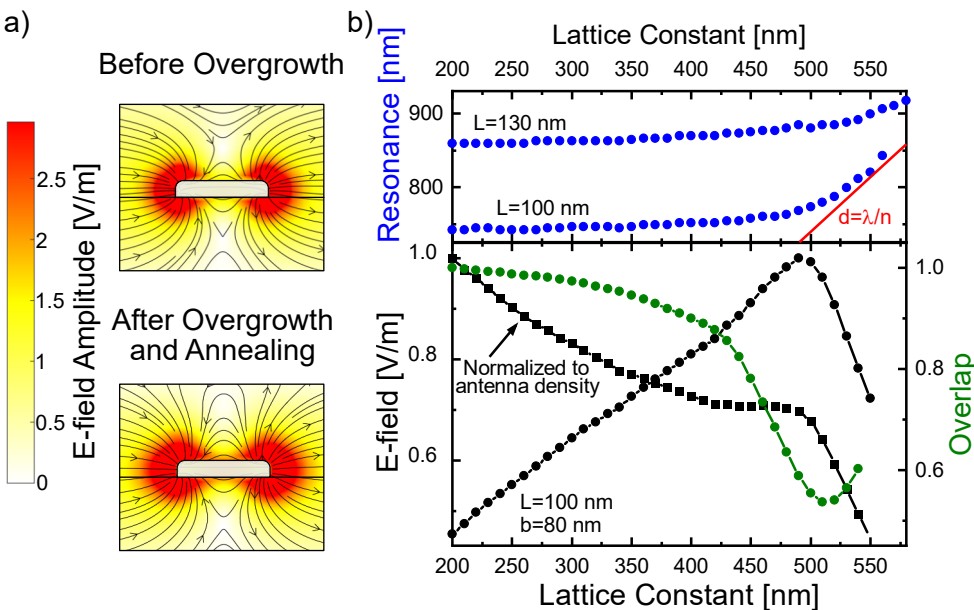

**Figure 2.** Theoretical results: (**a**) Calculated electric field strengths as colormaps including vector field for resonant excitation for non-overgrown and overgrown antennas with a width $w = 70$ nm, length $L = 110$ nm, thickness $h = 20$ nm, and lattice constant of $x = 400$ nm. (**b**) Blue: Resonant wavelength as a function of the antenna length. Red: Analytically calculated value for the lattice resonance. Black dots: Maximum electric field of an antenna with length $L = 100$ nm and width $w = 80$ nm. The highest electric field occurs for a lattice constant of 500 nm. Black squares: If normalized to the number of antennas in one array, the maximum electric field shifts to smaller lattice constants. Both black curves belong to the left axis. Green: Overlap between the electric field and the photoluminescence signal of the nanocrystals. The overlap is higher for smaller lattice constants.

## 4. Experimental Results

The antennas are characterized before overgrowth, after overgrowth, and after the high-temperature process in the tube furnace. A Fourier-transform infrared (FTIR) and a photoluminescence spectroscopy setup are used. As expected, the red shift of the resonance after overgrowth due to the higher refractive index of the surrounding medium and another red shift after annealing can be measured using FTIR in reflectance mode (see Figure 3).

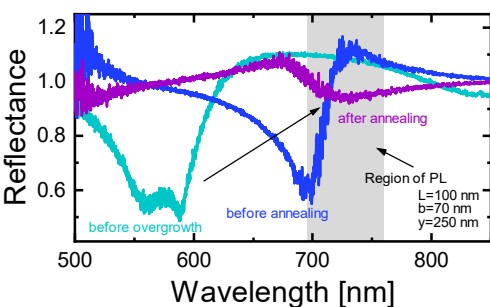

**Figure 3.** The resonance shifts to higher wavelengths after overgrowth and annealing. In the latter, the resonance is in the same range as the photoluminescence signal of the nanocrystals, so amplification can be expected after this fabrication step.

Photoluminescence is excited using a helium–cadmium laser with an emission wavelength of 325 nm and a power of 3 mW. Focusing is performed using a UV objective with 80× magnification on the sample in perpendicular geometry. The emitted radiation is directed through the same objective and through a beam splitter either onto a camera or into the spectrometer. The individual antenna fields can be seen in the camera, and L-shaped

markers are lithographed on the sides for better orientation. The laser spot has an extension of $d = 1.22 \cdot \frac{\lambda}{NA} \approx 721$ nm. Thus, an antenna field can be selectively excited. The nanocrystal layered system without an antenna field is measured as a reference $I_0$ in each case, as depicted in Figure 4a. Both spectra are subtracted from each other ($\Delta = I_A - I_0$). The results for different antenna fields before and after overgrowth and after annealing are shown in Figure 4b.

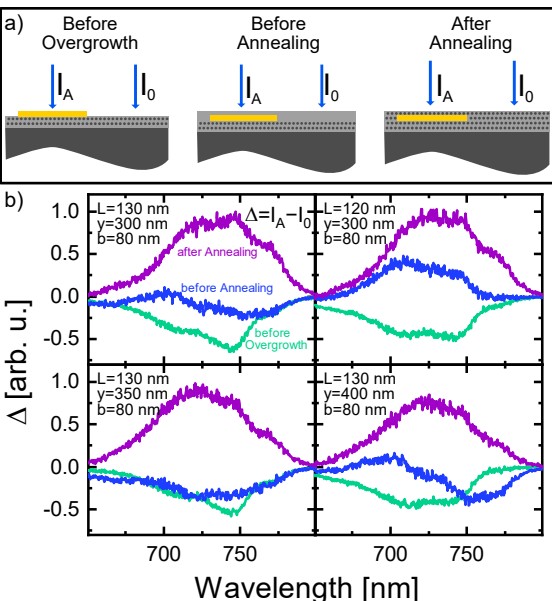

**Figure 4.** (**a**) The respective fabrication step for the measurement is displayed. For $I_A$, the signal of an antenna field is measured, and $I_0$ is the area without antennas. (**b**) This shows the difference between these measurements for the respective fabrication step. Before overgrowth, the PL signal is attenuated because the antennas act as centers for scattering. After overgrowth, there is no uniform attenuation or enhancement. After annealing, the signal is amplified for all fields.

Before overgrowth, the nanoantennas produce an attenuated signal in each case. Again, the resonance wavelengths are below the photoluminescence of the nanocrystals, so the overlap is very small. The nanoantannas act as non-resonant scattering centers that scatter the photoluminescence of the nanocrystals randomly in all directions. After overgrowth and before the high-temperature process, some attenuation and some enhancement is observed. After annealing, enhancement is evident in all fields shown, as the antennas' resonance shifts to the region of nanocrystal photoluminescence.

In the FTIR measurements, the red shift is not apparent for longer antennas or larger lattice constants, unlike in the theoretical calculations. In the measurements, the resonance seems to be rather blue-shifted. This is also matched by the overlap between the FTIR measurements and the PL spectrum of the nanocrystals (see Figure 5). The overlap is larger for smaller lattice constants. The largest obtained enhancements of photoluminescence by the nanocrystal are just above 6%. Apart from an outlier at a lattice constant of 250 nm, the gain of the PL signal decreases according to the overlap for larger lattice constants. Thus, here the influence due to the number of excited antennas is larger than the influence of the lattice resonance.

For the antenna length, a slight blue shift can also be observed in the FTIR measurements. This suggests that the sizes of the antennas are at the resolution limit of the lithography and that the antenna lengths between different antenna arrays do not differ much. As a trend, it can be summarized that the photoluminescence gain is greatest at smaller lattice constants. The 70 nm-wide antennas were so narrow that they did not adhere as well to the layered system, and many antennas were missing from the arrays. Due to the smaller number of antennas because of missing ones and the absence of grating

resonance, the gain here is much weaker, with a maximum of 1.7%. At this width, there is a general tendency for greater gain with smaller lattice constants or a greater number of antennas. At antenna widths of 80 nm, the photoluminescence can be enhanced by 6%, which means that the effect is much smaller compared to other approaches to enhance the photoluminescence [14]. One reason for this could be the different lifetimes in the nanoparticles and in the gold antennas. The lifetime in the nanocrystals depends on their size and is on the order of 60 µs [24]. The lifetime in the nanoantennas can be calculated from the linewidths of the simulation results and is in the order of a few femtoseconds. This significant difference may mean that little gain occurs through the antennas. Since the sizes of the nanoantennas are at the resolution limit of lithography and the length and width do not differ significantly, there is also no polarization dependence as would be expected for nanoantennas. Here, polarization dependence for the intensities of the PL signal of 0.4% is measured.

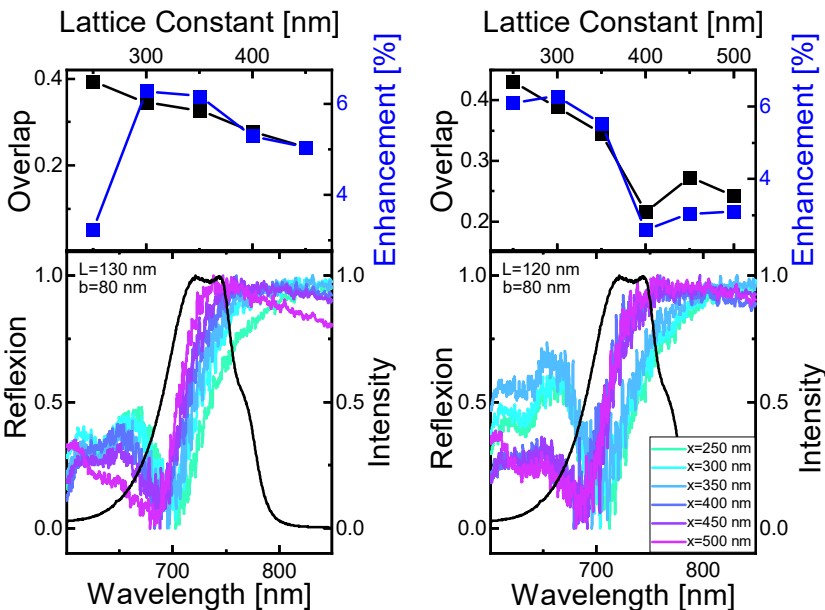

**Figure 5.** The experimental results for two antenna arrays are shown. In the lower part is the photoluminescence signal of the nanocrystals and the FTIR results depending on the lattice constant. The overlap decreases towards larger lattice constants. Accordingly, the gain shows a similar behavior. Except for one measurement point, the gain also decreases for larger lattice constants.

## 5. Conclusions

We have studied the manipulation of photoluminescence of silicon nanocrystals with gold antennas. We have demonstrated a reduction when the antennas are on top of the layered system with embedded nanocrystals because the non-resonant antennas act as centers for random scattering. Further, they are not resonant to the photoluminescence of the nanocrystals because the refractive index of the surrounding material is too low. After overgrowth and annealing of the antennas, all fields show a gain as the resonance wavelength shifts to the region of nanocrystal photoluminescence. The gain depends on the overlap of the photoluminescence signal with the resonance of the nanoantennas. This behavior can be confirmed. However, the gain is much lower than, for example, the SERS effect. This could be because the lifetime in the nanoantenna arrays is much shorter than in the nanocrystals. Thus, overgrowth of the antennas is a promising approach to amplify the photoluminescence signals. The gain should also be larger for similar lifetimes of the same order of magnitude. In addition, antenna fabrication becomes easier when longer wavelengths are to be amplified, as the antennas can then be correspondingly larger, which greatly simplifies fabrication.

**Supplementary Materials:** The following supporting information can be downloaded at: https://www.mdpi.com/article/10.3390/photonics9120985/s1, Figure S1: Theoretical resonances; Figure S2: Polarization dependence.

**Author Contributions:** Conceptualization, C.M., C.G. and R.K.; methodology, C.M., C.G. and R.K.; software, C.G. and R.K.; validation, C.M., C.G. and R.K.; formal analysis, C.G., R.K. and H.Q.; investigation, C.G., R.K. and H.Q.; resources, C.M.; data curation, C.G., R.K. and H.Q.; writing—original draft preparation, R.K.; writing—review and editing, R.K., C.G. and C.M.; visualization, R.K. and H.Q.; supervision, C.M.; project administration, C.G., R.K. and C.M.; funding acquisition, C.M. All authors have read and agreed to the published version of the manuscript.

**Funding:** The authors gratefully acknowledge financial support from the Deutsche Forschungsgemeinschaft (DFG) via TRR142 project B09.

**Institutional Review Board Statement:** Not applicable.

**Informed Consent Statement:** Not applicable

**Data Availability Statement:** Not applicable.

**Conflicts of Interest:** The authors declare no conflict of interest.

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
