# Peer review of "Influence of Gold Nanoantennas on the Photoluminescence of Silicon Nanocrystals"

_photonics, doi:10.3390/photonics9120985_

Round 1

Reviewer 1 Report

The manuscript presented by R. Köthemann and coauthors reports their study on the effect of gold nanoantennas on PL of SiNCs. The manuscript is well written and logically structured. Its topic is certainly of high interest to the readership of the journal. Therefore I recommand the publication of this manuscript after some minor revisions.

1. It might be a formatting problem in the downloaded version, but very often the units are missing in the text. This should be controlled.

2. On page 2 lines 63 & 64, authors claim that gold Nanoantennas show rounding up due to melting. Would it be possible to avoid this by reducing the temperature of the thermal processing step?

3. The manuscript will benefit from a schematic top view of the samples where also x and y directions are defined. 

4. Regarding figures 1c and 1d, I suggest including an image of a single antenna before overgrowth. 

5. On page 5, lines 143-145, the authors claim that missing antennas are the cause of reduced gain! How does the normalized gain (considering the actual number of antennas) compare?

Reviewer 2 Report

This manuscript reports on the influence of metal nanoantennas on the photoluminescence of silicon nanocrystals.

Here are some comments:

. In figure 2, are the maximum E-field in V/m or normalized?

. Most of the parameter values do not present the respective scales. The authors must inform if the parameter values are nanometers, micrometers, milliwatts, etc.

. In figure 2, it would be interesting adding a schematic of a portion of the array of nanoantennas with the corresponding dimensions. Also, a colormap of electric field distribution to highlight the field enhancement before and after overgrowth plus annealing. 

In the supplementary materials:

. It is important to inform the values of antenna length and distance between the antennas in the caption of Figure S1.

. Double-check typos in the first paragraph of the supplementary materials.
